# Study on Hydraulic Dampers Using a Foldable Inverted Spiral Origami Structure

Jingchao Guan [1,*] , Jingshun Zuo [1], Wei Zhao [2], Nobuyuki Gomi [1] and Xilu Zhao [1]

1   Department of Mechanical Engineering, Saitama Institute of Technology, Saitama 369-0293, Japan
2   Weichai Global Axis Technology Co., Ltd., Tokyo 107-0062, Japan
*   Correspondence: guanjingchao123@gmail.com

**Abstract:** Hydraulic dampers for the vibration damping of industrial machinery and building structures are typically cylindrical. This study proposes a novel, axially free-folding hydraulic damper of the origami type to improve the structural characteristics of the conventional cylinder shape with restricted effective stroke in relation to the overall length. First, the basic design equation of the proposed origami hydraulic damper was derived by demonstrating that the fold line cylinders on the sidewalls will always meet the foldable condition of the origami hydraulic damper, that is, $\alpha = \pi/n$ and $\pi/2n \leq \beta \leq \pi/n$. Next, the fluid flow characteristics inside the origami hydraulic damper and in the flow path were analyzed; it was determined that the actual damping force exerted on the origami damper was proportional to the square of the velocity of motion. Equations of motion were developed considering the derived damping force equation, and a vibration analysis method using the Range–Kutta numerical analysis technique was established. A validation test system with an origami hydraulic damper in a mass-spring vibration system was developed, and vibration tests were performed with actual seismic waves to verify the damping characteristics and effectiveness of the origami hydraulic damper. Furthermore, the orifice hole diameter at the end of the origami structure as well as the type of internal fluid, were varied in the vibration tests. The effect of the main components of the origami hydraulic damper on the damping effect was analyzed, revealing that the orifice hole diameter had a more significant effect than the internal fluid.

**Keywords:** origami hydraulic damper; viscous damping; origami engineering; vibration damping technology; vibration experiment; damping device

## 1. Introduction

Hydraulic dampers are primarily used as components for absorbing vibration and shock energy. Several studies have focused on their damping performance and the technical development of their application [1–5].

Hydraulic dampers have been investigated to enhance driver and passenger comfort by absorbing forces arising from road vibrations and collisions with obstacles [6–9]. Studies have demonstrated the effectiveness of installing semi-active hydraulic dampers along the diagonal direction of the rectangular frame of the main structure to mitigate earthquake-induced damage to buildings [10,11]. Magnetorheological (MR) dampers are developed by modifying the mechanical properties of conventional hydraulic dampers using an external magnetic field and have been the focus of numerous studies [12–16].

Nevertheless, the majority of existing hydraulic dampers are of the metal cylinder type, wherein the actual lengths of extension and contraction are limited compared to the total axial length, which limits their applicability when the installation space is limited. New techniques are required for weight reduction by replacing metal cylinders with dampers made of lightweight nonmetallic materials. Origami structures developed based on the desired mechanical performance of the structure in use have been investigated from fundamental perspectives such as geometry [17–22]. In some applications, tubular origami

structures that freely fold in the axial direction could replace conventional metal cylindrical dampers. The geometry of such foldable tubular origami structures and their deformation characteristics under the action of external forces have been investigated [23–28].

The lateral members mounted on either side of the engine along the front and rear of the vehicle play a significant role in absorbing collision energy in the case of a frontal collision. However, because of its slender thin-walled plate structure, it is prone to axial compression buckling, which can cause a sudden drop in collision energy absorption performance. To address this problem, the conventional thin-walled beam structure has been replaced by a tubular origami structure with a rectangular cross-section that can absorb a large amount of collision energy for the same weight [29–31].

Lightweight and flexible robots are sometimes required for robotic arms. Studies have used features of foldable tubular origami structures in the design of robot arms, hands, or grippers [32,33]. The movement of the origami structure can be controlled by exerting an external force on the interior or substructure of the foldable tubular structure, for which origami-shaped actuator findings have been developed [34–36].

However, the application of origami structures to hydraulic dampers using their free-folding characteristics has not yet been attempted. Origami hydraulic dampers are promising for applications involving a limited space, strict requirements for the total length and effective stroke of the dampers, and a need for weight reduction.

This study proposes a tubular origami hydraulic damper as an alternative to the conventional cylindrical damper. To solve the design problem of the origami hydraulic damper, the folding condition equation of the tubular origami structure was derived. Following the study of the configuration of the origami hydraulic damper, the damping force due to fluid flow within the origami structure was examined. Moreover, an experimental validation system using an origami hydraulic damper in a mass-spring vibration system was established, a numerical vibration analysis method using the Range–Kutta method was developed, and vibration experiments using actual seismic waves were conducted to validate the damping characteristics of the proposed origami hydraulic damper. To investigate this, vibration tests were conducted by altering the diameter of the damping hole at the end of the origami structure and using oil and water as the internal fluid. In addition, the effects of the main components of the paper-folding hydraulic damper on damping were investigated by evaluating and comparing the response displacement and acceleration of the moving body.

## 2. Materials and Methods

As shown in Figure 1, a conventional hydraulic cylinder damper is primarily composed of a cylinder, piston, and orifice hole. When the piston moves against the cylinder under the action of an external force, the hydraulic oil passes through the orifice hole of the piston, generating a damping force to counteract the relative motion.

In Figure 1, $L_{all}$ is the length of the hydraulic cylinder damper and $L_{work}$ is the distance the piston can traverse. When a hydraulic cylinder damper is used, the $L_{work}/L_{all}$ value indicates the percentage of available mounting space.

However, Figure 1 shows that the available $L_{work}/L_{all}$ value of the common hydraulic cylinder-type damper is less than 0.5. This is a drawback to its use in constricted mounting space.

In this scenario, instead of using a cylinder, if a new hydraulic damper was created using the folding function of the inverted spiral origami structure, the available $L_{work}/L_{all}$ value of the damper, as shown in Figure 2, would be significantly enhanced.

Nonetheless, it is essential to examine the configuration parameters of the inverted spiral origami structure shown in Figure 2 to evaluate the conditions under which it can be folded freely to use as a hydraulic damper.

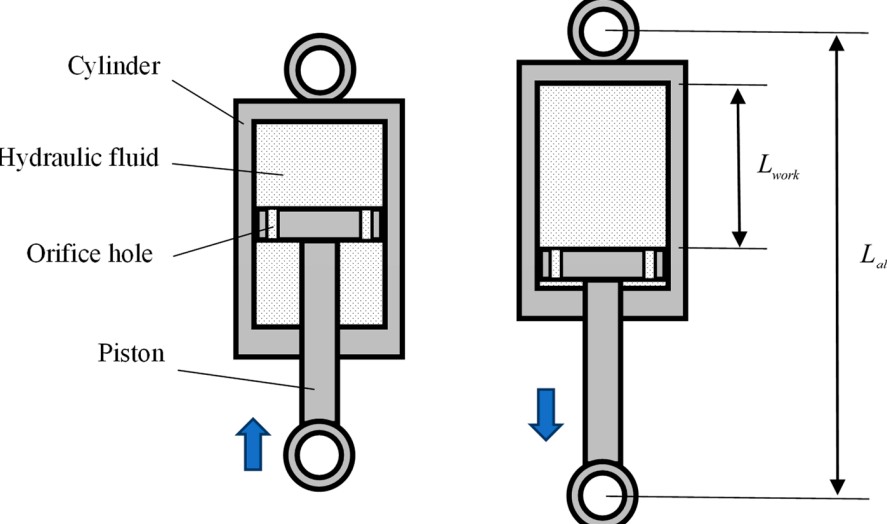

**Figure 1.** Conventional hydraulic cylinder damper.

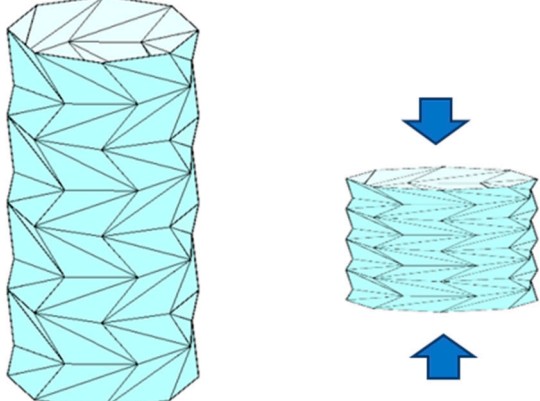

**Figure 2.** Inverted spiral origami structure.

### 2.1. Inverted Spiral Origami Structure

As shown in Figure 3, assuming six-fold lines *oa*, *ob*, *oc*, *od*, *oe*, and F drawn from the center point of a sheet of paper, its solid line represents a mountain fold, and its dotted line represents a valley fold. The angles composed by each fold line are $\theta_1$, $\theta_2$, $\theta_3$, $\theta_4$, $\theta_5$, and $\theta_6$. The requirement for the paper to be folded flat is expressed as follows [37].

$$\theta_1 + \theta_3 + \theta_5 = \theta_2 + \theta_4 + \theta_6 \tag{1}$$

Thus, the prerequisites for flattening paper must always be satisfied. Folding can be performed by alternating the left and the right fold, maintaining an identical sum of the angles of the left and right folds.

Figure 4 shows an example of an inverted spiral origami structure with an unfolded view on the left and a cylindrical origami structure close to the circumference on the right. Each node in Figure 4 consists of a node and six folds. *n* denotes the number of segments in the origami structure.

To satisfy the requirement of flat folding along the axial direction, it is necessary to consider how to identify the inclination angles *α* and *β* of the fold based on the number of segments *n* of the origami structure, as shown in Figure 4.

Figure 5a shows a drawing of the inverted spiral origami structure depicted in Figure 4, with only one section extracted and unfolded, wherein $x_0$ represents the direction of the original axis of the strip of flat paper.

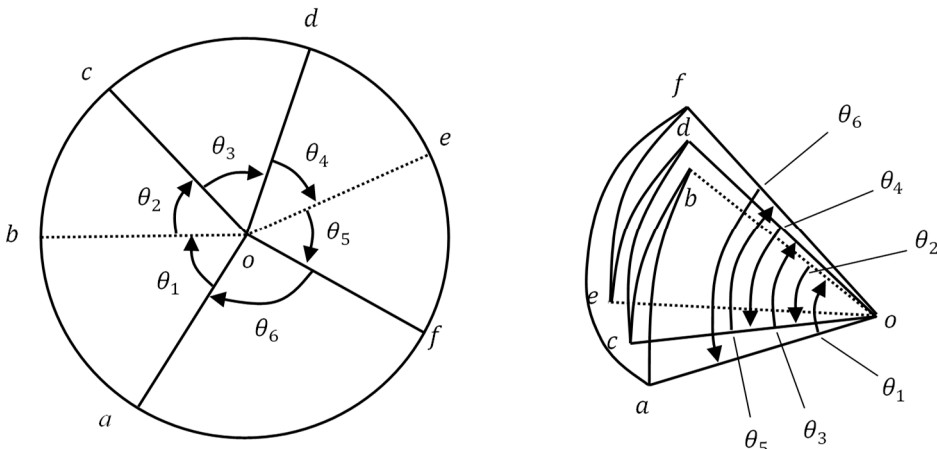

**Figure 3.** Concept of flat folding a sheet of paper by one point and six-fold lines.

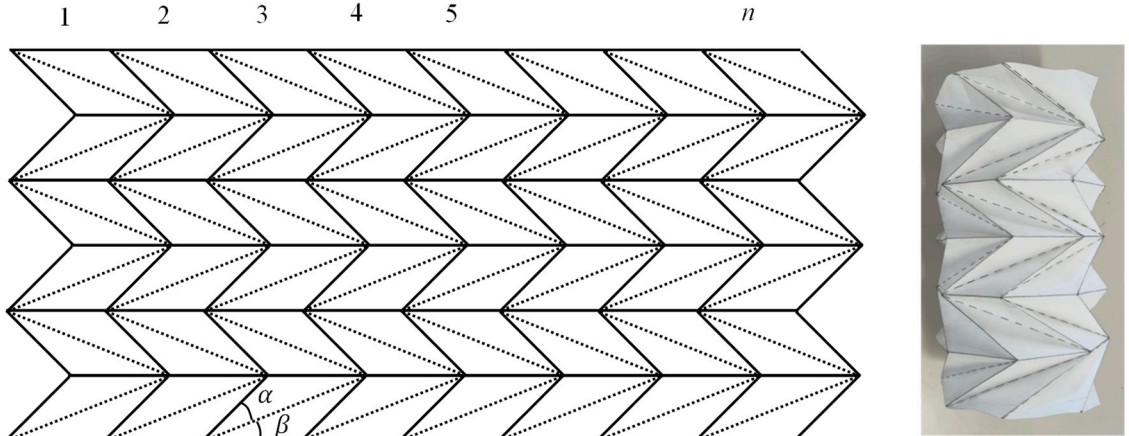

**Figure 4.** One-point and six-fold line-type cylindrical origami structure.

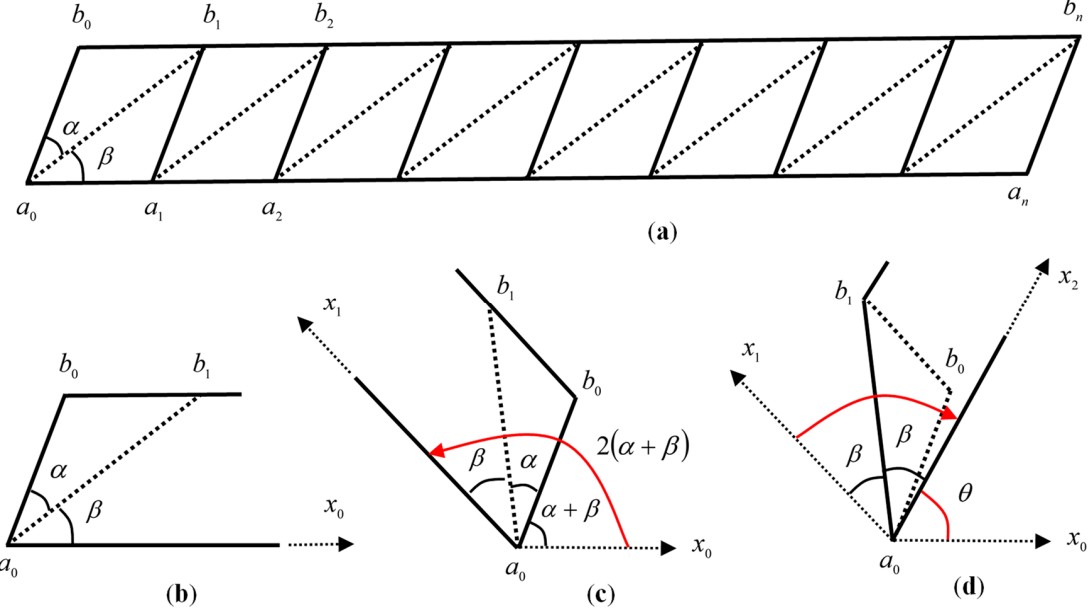

**Figure 5.** Condition for closing the one-point and six-fold line type cylindrical origami structure. (**a**) One section of the unfolded origami. (**b**) One unit of the unfolded origami. (**c**) First folding along angle ($\alpha + \beta$) in plus direction. (**d**) Second folding along angle ($\beta$) in minus direction.

First, a mountain-shaped fold was performed to fold the strip of flat paper into a cylinder, using $a_0b_0$ of the left-most section shown in Figure 5b as the folding axis, and the original axis direction $x_0$ was rotated by an angle $2(\alpha + \beta)$ in the positive direction, as shown in Figure 5c, to acquire a new axis direction $x_1$.

Next, the axial direction $x_1$ was rotated in the negative direction at the angle of $2\beta$, with $a_0b_1$ as the fold axis and the valley fold in the opposite direction, as shown in Figure 5d, resulting in a new axial direction of $x_2$. Because of the two folds of the left-most segment, the original angle of rotation of the flat strip of paper in the axial direction $x_0$ is expressed by the following equation.

$$\theta = 2(\alpha + \beta) - 2\beta = 2\alpha \tag{2}$$

The same origami operation is repeated for the number of segments $n$ from the left to the right, such that the total angle $2n\alpha = 2\pi$ with the original axial direction $x_0$ is folded to close the circumference after one round. The equation for the condition of closing the circumference is presented below.

$$\alpha = \frac{\pi}{n}. \tag{3}$$

In contrast, the circumferentially closed origami structure shown in Figure 6a is folded flat when compressed along the axial direction, as shown in Figure 6b; the conditions satisfied by angle $\beta$ must be considered.

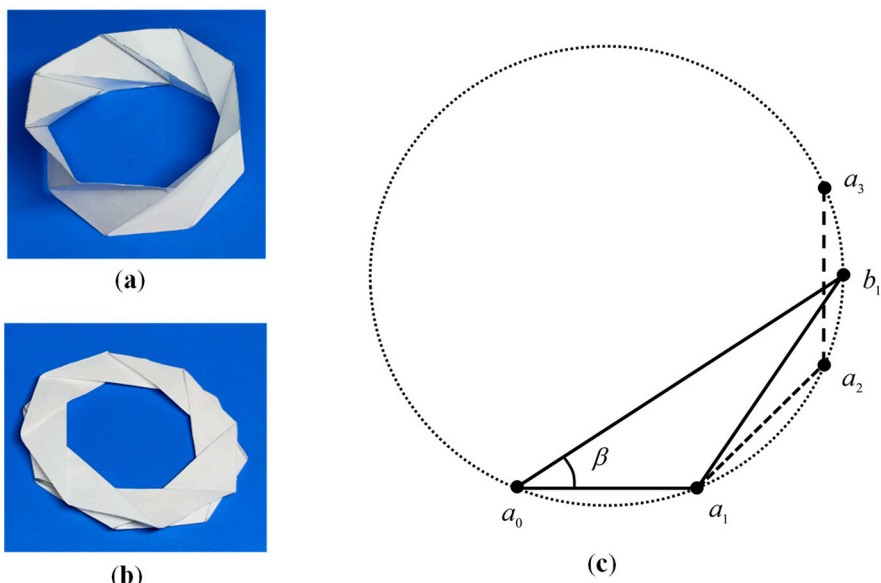

(a)

(b)

(c)

**Figure 6.** Folding condition for the one-point and six-fold line type cylindrical origami structure. (**a**) Origami circumferentially closed along the axial direction. (**b**) Folded flat Origami state. (**c**) Flattened states schematic diagram.

In this case, a triangle $a_0a_1b_1$ was extracted, and its flattened and folded states are shown in Figure 6c. Because the nodes $a_0$, $a_1$, and $a_2$ are located on the circumference, shown by the dotted line, resulting in a regular $n$ angle, and the node $b_1$ in the flat folded state is located between arcs $a_2a_3$, the conditions satisfied by angle $\beta$ can be expressed as follows, considering that the arc $a_1a_2$ is $1/n$ in the circumference.

$$\frac{\pi}{2n} \le \beta \le \frac{\pi}{n}. \tag{4}$$

Once the number of segments $n$ is established for the inverted helical origami structure that will be used in the hydraulic damper, the angles $\alpha$ and $\beta$ calculated using Equations (3) and (4) can be used to generate the inverted spiral origami structure, as shown in Figure 4.

In this study, assuming that node $b_1$ lies in the center of arc $a_1 a_2$, angle $\beta$ is represented by the following equation.

$$\beta = \frac{3\pi}{2n}. \tag{5}$$

### 2.2. Inverted Spiral Origami Type Hydraulic Damper

An experimental setup, shown in Figure 7, was developed to study the performance of an inverted spiral origami-type hydraulic damper. As shown in the figure, the experimental setup was composed of an origami damper, elastic spring, mass block, frame, fixed plate, moving plate, and oil tube.

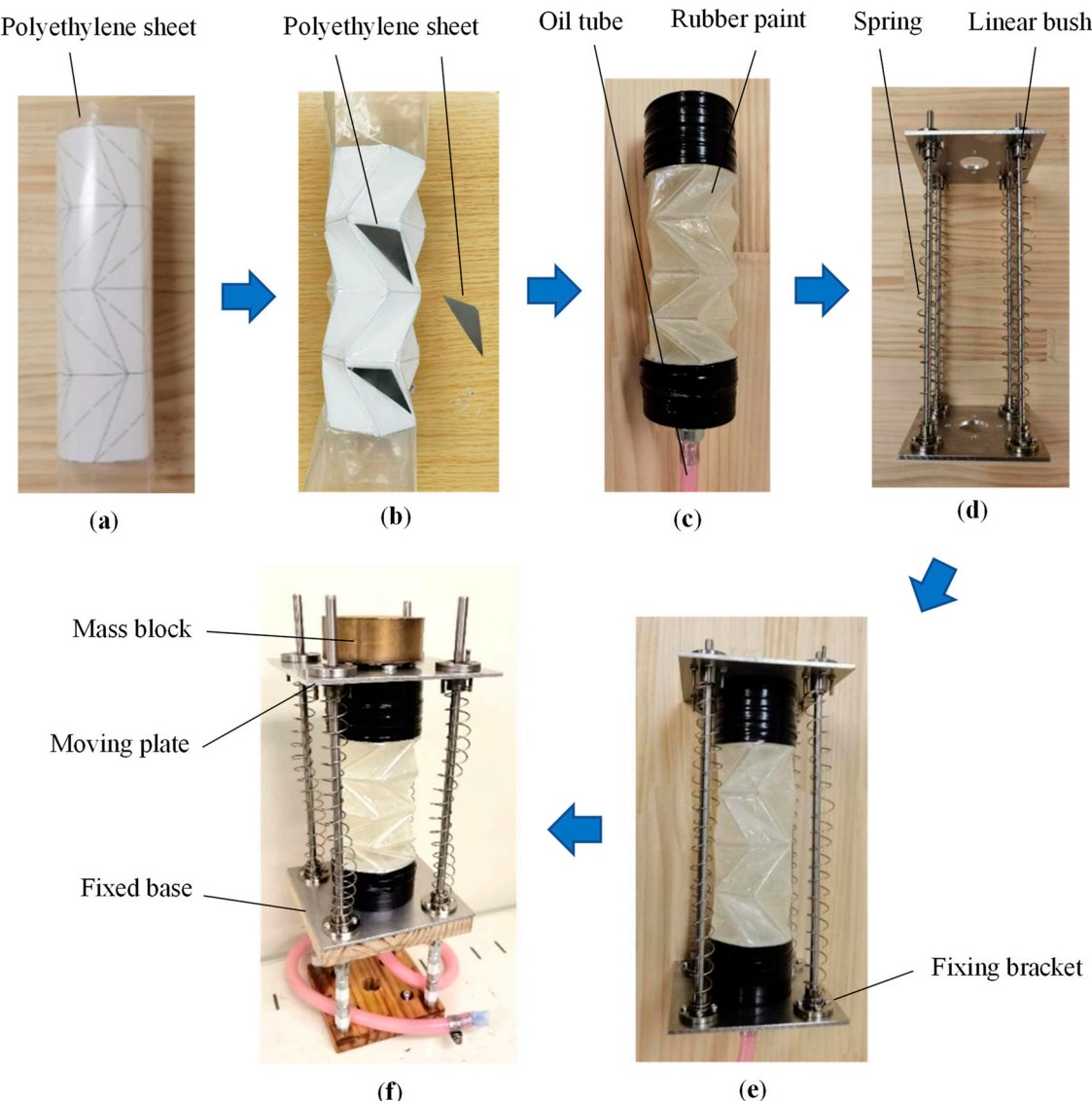

**Figure 7.** Proposed inverted spiral origami hydraulic damper and main components. (**a**) Polyethylene cylindrical sheet. (**b**) Thin triangular metal sheet. (**c**) End faces of origami damper. (**d**) Spring frame. (**e**) Origami damper. (**f**) Vibration test device.

First, as shown in Figure 7a, an adhesive was used to attach a transparent polyethylene cylindrical sheet with the paper printed on the results of the Auto CAD 2021 design.

Subsequently, as shown in Figure 7b, it was folded along the printed lines to form an inverted spiral origami structure. A thin triangular metal plate was glued to each side of the structure to enhance its rigidity.

The inner and outer surfaces of the origami structure were covered with liquid rubber, as shown in Figure 7c, and one end face was completely sealed to prevent the oil inside from leaking out. The other end face was used as the orifice hole, and an oil tube was attached to create an origami hydraulic damper set.

The assembled hydraulic damper set was mounted on a frame with four elastic springs, as shown in Figure 7d, forming a vibration model with the combined elastic springs and hydraulic dampers, as shown in Figure 7e.

Finally, a fixed base was mounted on the lower end of the assembled vibration model, and a mass block was mounted on the upper end, forming the experimental setup to investigate the performance of the inverted spiral origami hydraulic damper.

In experiments, the mass block vibrates up and down in the hydraulic oil inside the folded paper hydraulic damper. The hydraulic oil enters and exits through an orifice hole at the lower end of the origami damper, creating a vibration-damping force.

### 2.3. Damping Force of Origami Type Hydraulic Damper

The damping effect of the origami hydraulic damper, shown in Figure 7, consists primarily of the resistance produced by the internal hydraulic oil as it flows through the orifice hole at the lower end of the origami hydraulic damper and the oil tube.

A simplified analytical model of the origami hydraulic damper, oil pipe, etc., is shown in Figure 8, where $x$ is the height of the origami hydraulic damper, corresponding to the actual working distance of the hydraulic damper. $d$ represents the diameter of the center of the origami hydraulic damper, $d_1$ indicates the diameter of the orifice hole of the origami hydraulic damper flowing to the oil pipe, and $d_2$ represents the diameter of the oil tube. $P$ represents the pressure at the top of the origami damper, $P_c$ indicated the pressure at the bottom orifice hole of the origami hydraulic damper, $v$ shoes the flow rate of the hydraulic oil inside the origami hydraulic damper, $P_1$ and $v_1$ represent the pressure and flow rate at the outlet of the orifice, and $P_2$ and $v_2$ indicate the pressure and flow rate at the end of the oil tube.

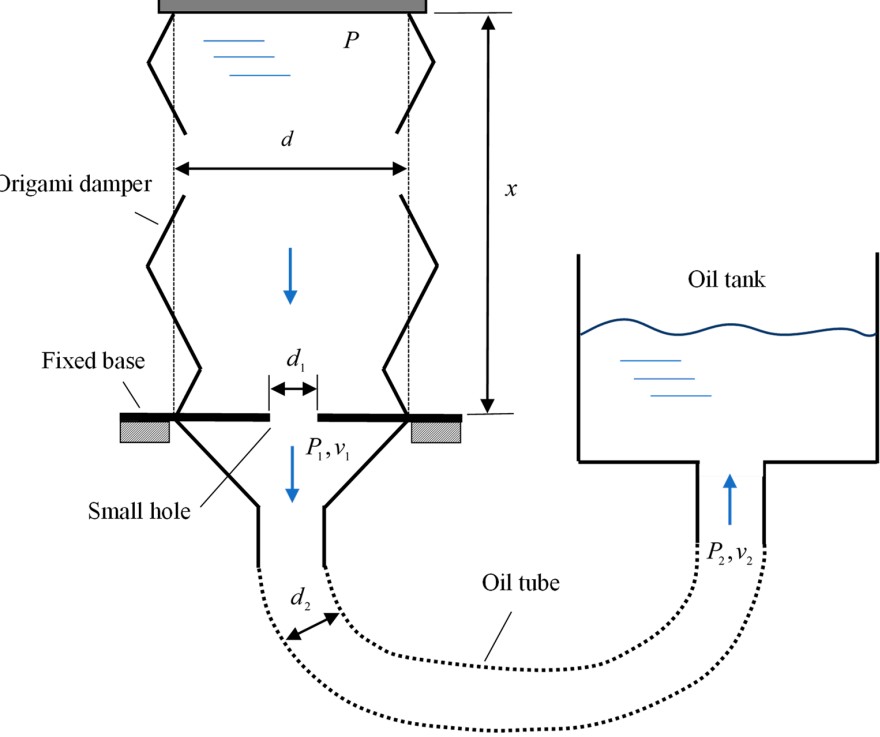

**Figure 8.** Analysis model of an inverted spiral origami-type hydraulic damper.

The flow rate of the hydraulic fluid through the orifice hole in the body of the origami hydraulic damper can be derived using the following equation [38]:

$$Q = CA_1 \sqrt{\frac{2}{\rho}(P - P_1)},\tag{6}$$

where $C$ is the flow coefficient, $A_1$ is the cross-sectional area of the orifice hole, and $\rho$ is the density of the hydraulic fluid. Considering the relationship $Q = v_1 A_1$ between flow quantity and flow rate, the internal pressure of the origami hydraulic damper can be formulated as follows:

$$P = P_1 + \frac{\rho}{2C^2}v_1^2.\tag{7}$$

The pressure loss of the hydraulic fluid as it flows through the tube can be computed using the following Darcy–Weisbach equation [38].

$$P_1 - P_2 = \lambda \frac{\rho v_2^2}{2} \frac{L}{d_2},\tag{8}$$

where $\lambda$ is the friction coefficient of the oil pipe, and $L$ is the length of the oil tube. The pressure loss due to the rapid expansion of the cross-sectional area when the hydraulic fluid enters the tank from the oil tube can be evaluated using the following equation [38]:

$$P_2 = K_c \frac{\rho v_2^2}{2},\tag{9}$$

where, $K_c$ is the loss coefficient.

By rearranging Equations (7)–(9), the following equation can be derived.

$$P = \frac{\rho}{2c^2}v_1^2 + \frac{\lambda \rho L}{2d_2}v_2^2 + \frac{K_c \rho}{2}v_2^2.\tag{10}$$

As the relationship between the flow velocities of each part of the origami hydraulic damper is equal, the relationship between the velocity $\dot{x}$ at its upper-end face and the internal flow velocity can be expressed as follows.

$$v_1 = \frac{d^2}{d_1^2}\dot{x},\tag{11}$$

$$v_2 = \frac{d^2}{d_2^2}\dot{x}.\tag{12}$$

Substituting Equations (11) and (12) into Equation (10), the internal pressure exerted on the upper surface of the hydraulic damper of the origami can be expressed as

$$P = \frac{\rho d^4}{2}\left(\frac{1}{c^2 d_1^4} + \frac{\lambda L}{d_2^5} + \frac{K_c}{d_2^4}\right)\dot{x}^2.\tag{13}$$

Furthermore, by multiplying the cross-sectional area of the origami hydraulic damper with Equation (13), the forces exerted on the origami hydraulic damper in actual use can be expressed as follows.

$$F_{damper} = \frac{\pi \rho d^6}{8}\left(\frac{1}{c^2 d_1^4} + \frac{\lambda L}{d_2^5} + \frac{K_c}{d_2^4}\right)\dot{x}^2.\tag{14}$$

Equation (14) shows a nonlinear relationship between the force $F_{damper}$ of the origami hydraulic damper and velocity $\dot{x}$. The coefficient part of Equation (14) is defined as the new origami hydraulic damper coefficient, as expressed in the following equation.

$$c_{damper} = \frac{\pi \rho d^6}{8} \left( \frac{1}{c^2 d_1^4} + \frac{\lambda L}{d_2^5} + \frac{K_c}{d_2^4} \right). \tag{15}$$

The acting force of the origami hydraulic damper can be expressed as follows.

$$F_{damper} = c_{dampaer} \dot{x}^2. \tag{16}$$

Equation (16) can be applied to the analysis of a vibration system employing an origami hydraulic damper. Figure 9 shows the relationship between the acting force and velocity of the proposed origami hydraulic damper with the configuration parameters listed in Table 1 calculated using Equations (15) and (16).

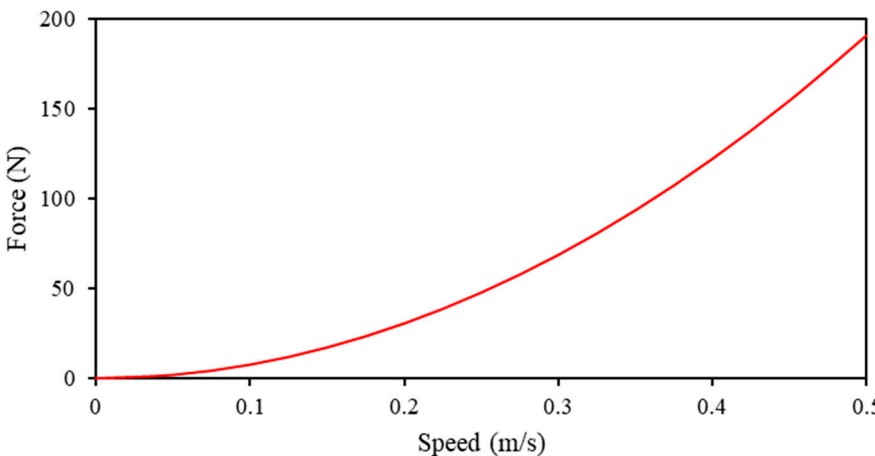

**Figure 9.** Relationship between the applied force and speed of the cylindrical origami hydraulic damper.

**Table 1.** Examples of configuration parameters for cylindrical origami hydraulic dampers.

| Items | Parameters |
|---|---|
| Hydraulic oil density $\rho$ | 860 kg/m$^3$ |
| Orifice flow coefficient $c$ | 0.61 |
| Coefficient of friction of oil tube $\lambda$ | 0.005 |
| Pressure drop coefficient of rapid expansion tube $K_c$ | 0.65 |
| Oil tube length $L$ | 300 mm |
| Average diameter of cylindrical origami hydraulic damper $d$ | 50 mm |
| Orifice hole diameter $d_1$ | 12 mm |
| Oil tube diameter $d_2$ | 15 mm |

When the origami hydraulic damper is used in a vibration model, as shown in Figure 7, the velocity value is considered relatively small as it continues to oscillate and deform, alternating between positive and negative values, with zero velocity as the center point. However, Figure 9 shows a nonlinear relationship between the force and velocity of the origami hydraulic damper.

### 2.4. Vibration Analysis Method

The dynamic analysis model used to study the damping effect of the origami-type hydraulic damper (Figure 7) is illustrated in Figure 10, where $m$ is the mass of the mass block, $c$ is the damping force due to the friction effect, $F_{dampaer}$ is the force of the origami-type hydraulic damper expressed by Equation (14), and $\ddot{x}_e$ represents the acceleration of the external excitation signal.

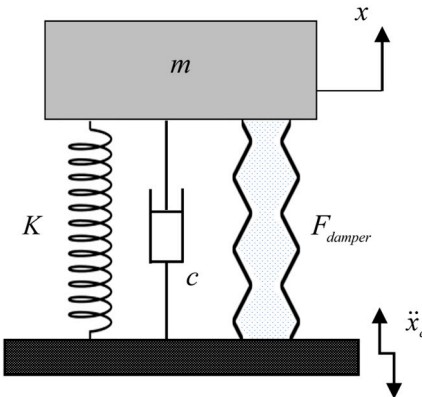

**Figure 10.** Analytical model for examining the damping effect of an inverted spiral origami hydraulic damper.

Considering the forces acting on the hydraulic damper for paper folding, the equations of motion can be described as follows:

$$m\ddot{x} + c\dot{x} + c_{damper}\dot{x}^2 + Kx = -m\ddot{x}_e, \tag{17}$$

where $c_{damper}$ represents the origami hydraulic damper coefficient provided by Equation (15), dictated by the configuration parameters of the origami hydraulic damper.

The equation of motion (16) for a vibrating system with an origami-type hydraulic damper incorporates a squared term for the velocity, suggesting a nonlinear differential equation.

For vibration analysis, the equation of motion (16) is reformulated as the Runge–Kutta standard equation, as follows.

$$\frac{d\dot{x}}{dt} = -\frac{c}{m}\dot{x} - \frac{c_{damper}}{m}\dot{x}^2 - \frac{K}{m}x - \ddot{x}_e, \tag{18}$$

$$\frac{dx}{dt} = \dot{x}. \tag{19}$$

At the time step $t_i$, using $x_i$, $\dot{x}_i$, $\dot{x}_{ei}$, and the time step length $\Delta t$, applying the Runge–Kutta iterative analysis method to Equations (18) and (19) leads to the solution of the vibrational displacement and velocity of the time series, as follows.

$$\Delta\dot{x}_1 = \Delta t\left(-\frac{c}{m}\dot{x}_i - \frac{c_{damper}}{m}\dot{x}_i^2 - \frac{k}{m}x_i - \ddot{x}_{ei}\right), \tag{20}$$

$$\Delta x_1 = \Delta t\dot{x}_i, \tag{21}$$

$$\Delta\dot{x}_2 = \Delta t\left[-\frac{c}{m}\left(\dot{x}_i + \frac{\Delta\dot{x}_1}{2}\right) - \frac{c_{damper}}{m}\left(\dot{x}_i + \frac{\Delta\dot{x}_1}{2}\right)^2 - \frac{k}{m}\left(x_i + \frac{\Delta\dot{x}_1}{2}\right) - \ddot{x}_{ei}\right], \tag{22}$$

$$\Delta x_2 = \Delta t\left(\dot{x}_i + \frac{\Delta\dot{x}_1}{2}\right), \tag{23}$$

$$\Delta \dot{x}_3 = \Delta t \left[ -\frac{c}{m}\left( \dot{x}_i + \frac{\Delta \dot{x}_2}{2} \right) - \frac{c_{damper}}{m}\left( \dot{x}_i + \frac{\Delta \dot{x}_2}{2} \right)^2 - \frac{k}{m}\left( x_i + \frac{\Delta x_2}{2} \right) - \ddot{x}_{ei} \right], \tag{24}$$

$$\Delta x_3 = \Delta t \left( \dot{x}_i + \frac{\Delta \dot{x}_2}{2} \right), \tag{25}$$

$$\Delta \dot{x}_4 = \Delta t \left[ -\frac{c}{m}\left( \dot{x}_i + \Delta \dot{x}_3 \right) - \frac{c_{damper}}{m}\left( \dot{x}_i + \Delta \dot{x}_3 \right)^2 - \frac{k}{m}\left( x_i + \Delta x_3 \right) - \ddot{x}_{ei} \right], \tag{26}$$

$$\Delta x_4 = \Delta t \left( \dot{x}_i + \Delta \dot{x}_3 \right). \tag{27}$$

The new time-step solution can be updated as follows.

$$t_{i+1} = t_i + \Delta t, \tag{28}$$

$$x_{i+1} = x_i + \frac{1}{6}(\Delta x_1 + 2\Delta x_2 + 2\Delta x_3 + \Delta x_4), \tag{29}$$

$$\dot{x}_{i+1} = \dot{x}_i + \frac{1}{6}(\Delta \dot{x}_1 + 2\Delta \dot{x}_2 + 2\Delta \dot{x}_3 + \Delta \dot{x}_4), \tag{30}$$

By repeating the above calculation procedure, the following solution is obtained.

$$x_0, x_1, x_2, , , , , , , x_n. \tag{31}$$

### 2.5. Vibration Verification Experiment

The vibration test system designed to verify the damping effect of the hydraulic dampers for origami studied in the previous section is composed of a shaker, amplifier, signal generator, laser displacement meter, accelerometer, fast Fourier transform (FFT) analyzer, and PC for processing the results, as shown in Figure 11.

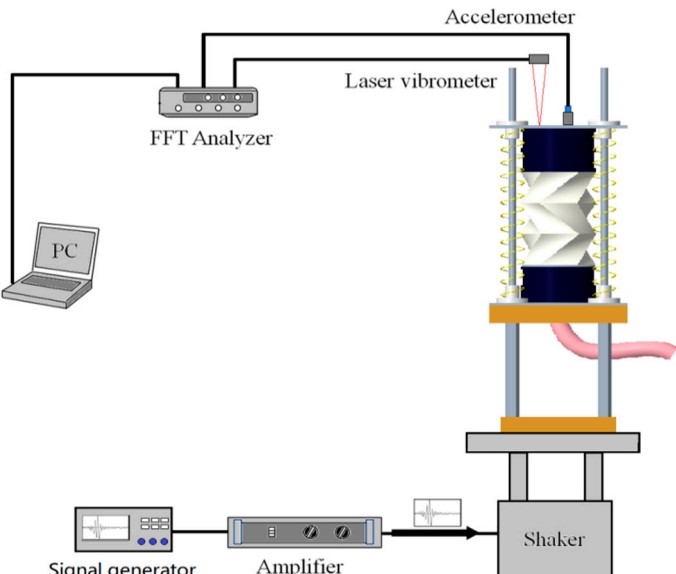

**Figure 11.** Origami hydraulic damper damping effect measurement experiment flow chart.

The actual measurements were performed as detailed in the flow chart (Figure 11). The shaker of the origami hydraulic damper vibrates because of a vibration signal that is amplified and supplied from the signal generator to the shaker.

The displacements of the bottom fixed base and top-end surface of the origami hydraulic damper were detected using a laser displacement meter and recorded by a data logger. Meanwhile, the acceleration signal of the top-end surface of the origami hydraulic

damper was measured by an accelerometer and recorded by an FFT analyzer. Finally, the measured vibration response results were processed and delivered using a computer.

An image of the actual experimental equipment developed is shown in Figure 12; the specific parameters of each component of the experimental equipment and the measurement instruments are listed in Table 2.

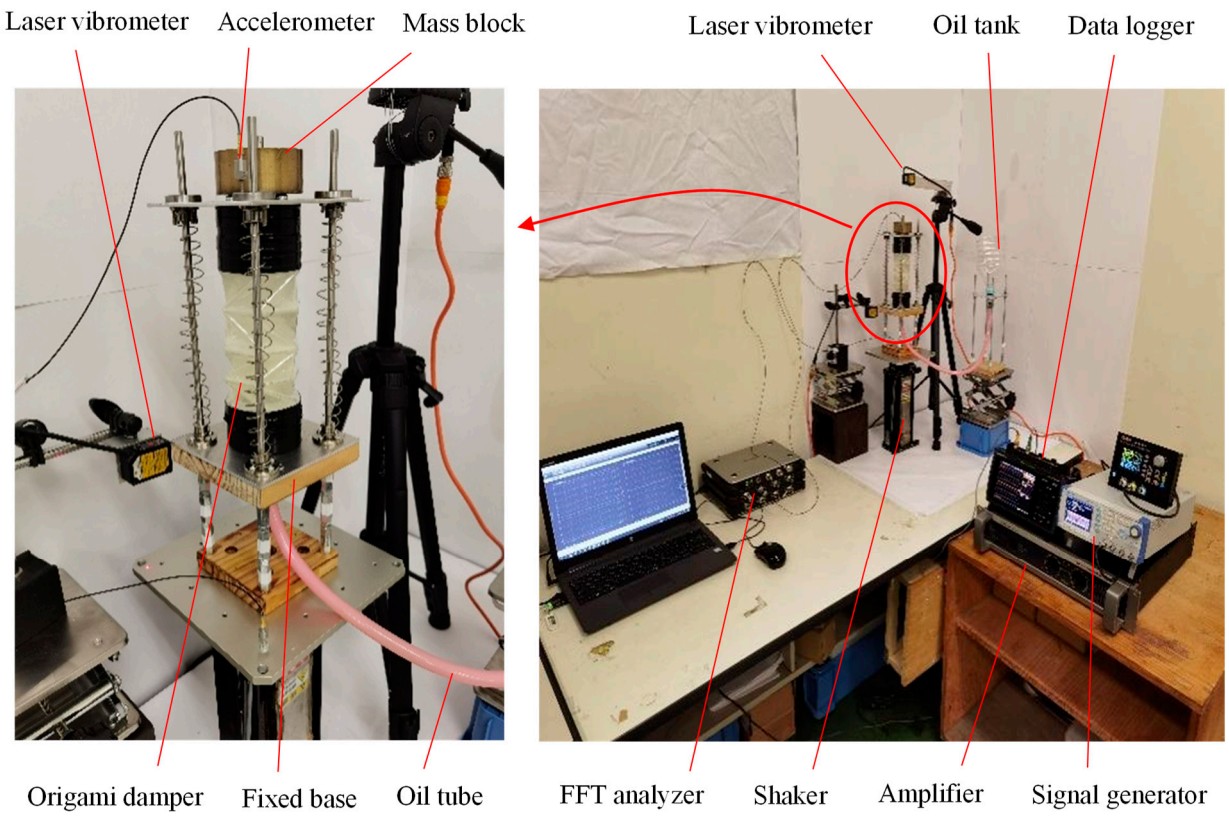

**Figure 12.** Vibration experiment system to confirm the damping effect of origami hydraulic damper.

**Table 2.** Detailed parameters of the experiment system of the origami hydraulic damper.

| Items | Parameters |
|---|---|
| Spring length $l_0$ | 110 mm |
| Spring coefficient $K$ | 280 N/m |
| Mass of mass block $m$ | 1.431 kg |
| Friction damping coefficient $c$ | 0.35 Ns/m |
| FFT analyzer | Shimadzu Corporation DS-3000 |
| Laser vibrometer | Optex FA CD100 $\pm$ 50 |
| Accelerometer | Onosokki NP-3421 |
| Shaker | SAN-Esu SSV-60S |
| Signal generator | NF Corporation WF1973 |
| Data logger | HIOKI Corporation LR8431 |

Experiments and numerical analyses were performed with the vibration model of the origami hydraulic damper shown in Figure 7 using a sinusoidal signal (frequency 4 Hz). The results of the vibration displacements are shown in Figure 13.

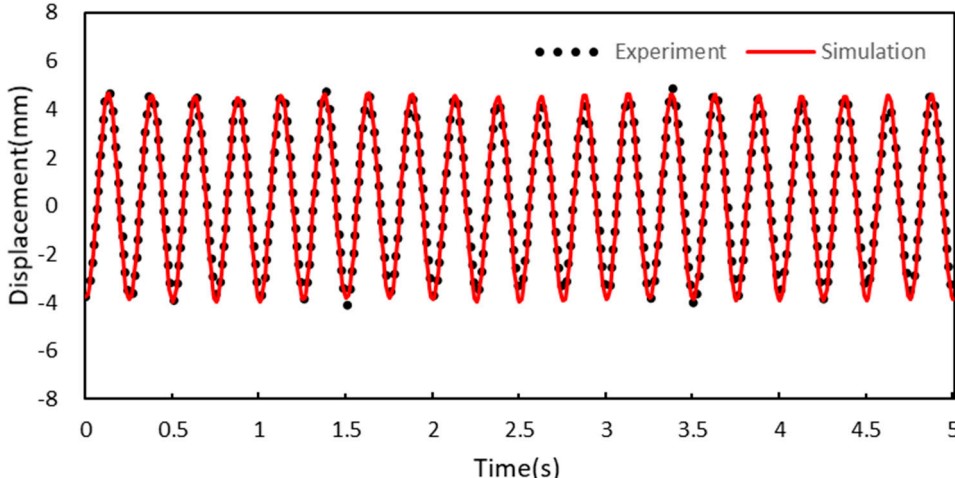

**Figure 13.** Comparison of vibration displacement measurement experiment and numerical analysis results.

## 3. Results and Discussion

Figure 12 shows the equipment used to test the damping effect of origami hydraulic dampers in a complex vibration environment using actual seismic wave excitation. The vibration experiments were conducted using two scenarios, one with and one without hydraulic oil in the origami damper. The results of the response displacement and response acceleration of the mass block were compared. The acceleration waveforms of the seismic waves used in the shaker experiments are shown in Figures 14 and 15, respectively. The acceleration waveforms of the EI Centro NS seismic waves are shown in Figure 14, and those of the Taft seismic waves are displayed in Figure 15.

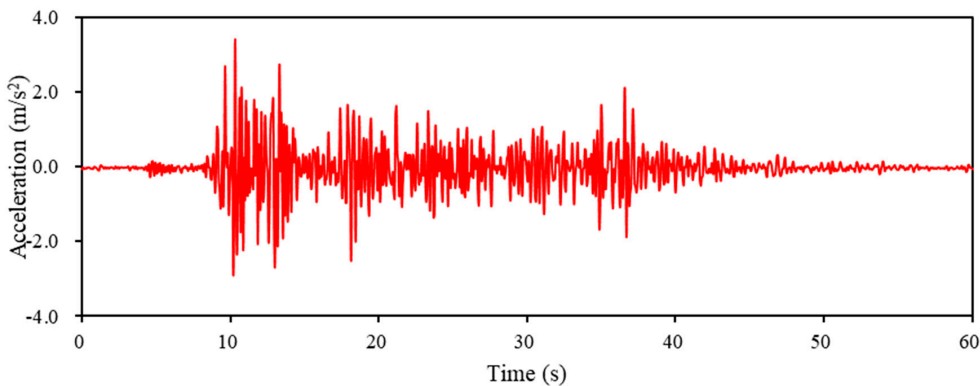

**Figure 14.** Acceleration waveform of EI Centro NS seismic wave used for excitation experiment.

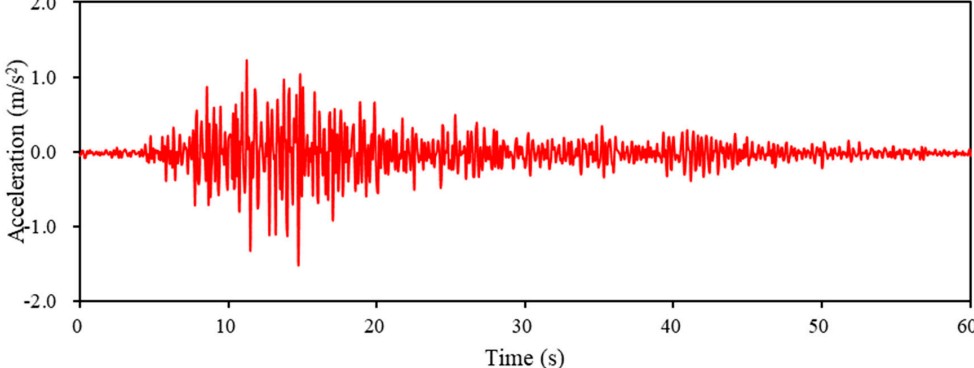

**Figure 15.** Acceleration waveform of Taft seismic wave used for excitation experiment.

### 3.1. Vibration Verification Experiment

Figure 16 shows the results when shaking was measured using the EI Centro NS seismic waves. Figure 16a illustrates the response displacement of the time series; Figure 16b illustrates the results of the spectral analysis of the response displacement in the frequency domain; Figure 16c illustrates the response acceleration of the time series; Figure 16d illustrates the results of the spectral analysis of the response acceleration in the frequency domain. The solid blue line indicates the measurement results without hydraulic oil, whereas the red dotted line indicates the measurement results with hydraulic oil.

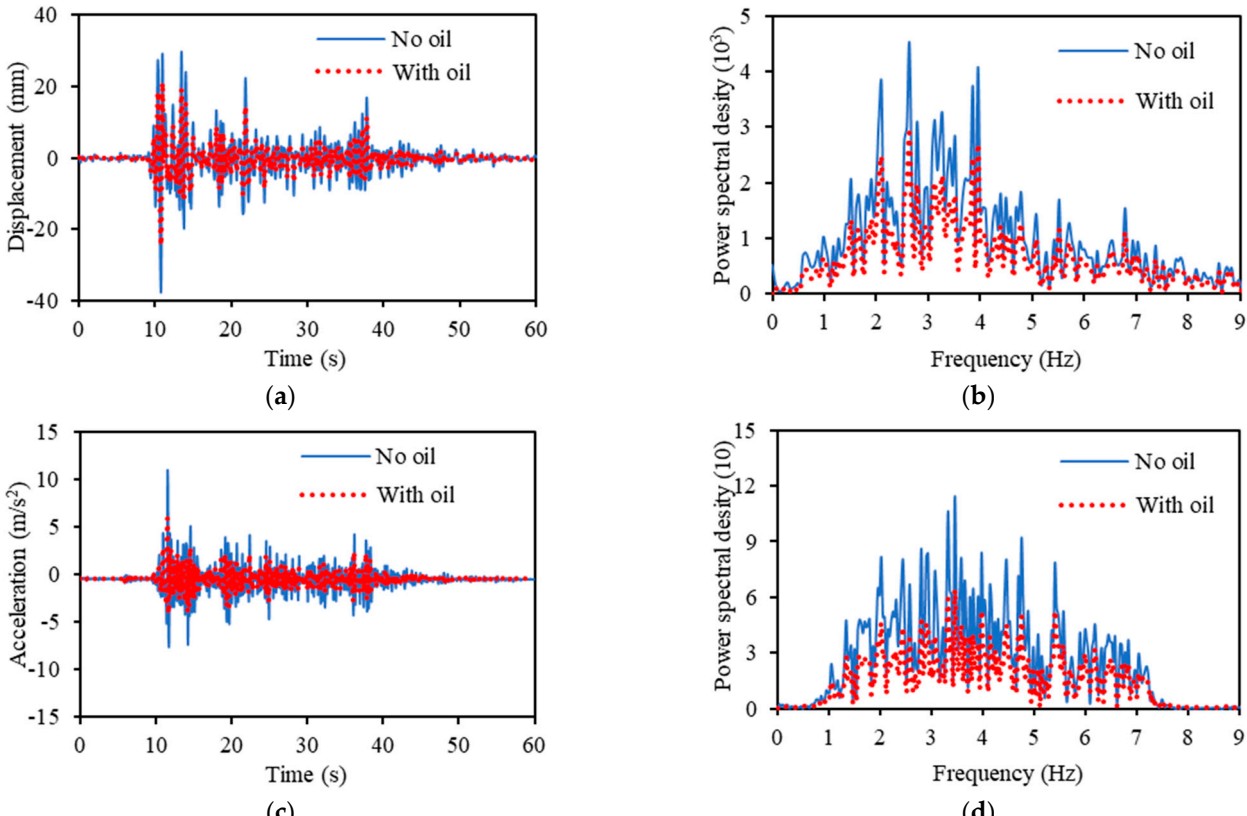

**Figure 16.** Measurement results when excited by EI Centro NS seismic wave. (**a**) Response displacement. (**b**) Spectral distribution of displacement. (**c**) Response acceleration. (**d**) Spectral distribution of acceleration.

Figure 16 indicates that the response displacement and acceleration of the scenario wherein hydraulic fluid is included are apparently smaller than the response displacement and acceleration of the scenario in which no hydraulic fluid is included when vibrations are applied using EI Centro NS seismic waves. The results of the spectrum analysis indicate that the addition of the hydraulic fluid reduces the vibration response over a wide range of frequencies and offers a damping effect.

The results for the response displacement in Figure 16a and response acceleration in Figure 16c are summarized in Table 3. The maximum value of the response displacement was reduced by 53.33%, and the average was reduced by 55.78%. The maximum value of the response acceleration was reduced by 87.07%, and the average was reduced by 39.05%. The kurtosis value of the response displacement was increased by 2.18%, and the skewness value was increased by 0.98%. The kurtosis value of the response acceleration was reduced by 56.73%, and the skewness value was reduced by 26.39%.

**Table 3.** Comparison of measurement results when excited by EI Centro NS seismic wave.

|  | | Displacement | | Acceleration |
|---|---|---|---|---|
| **Maximum value** | No oil | 37.72 | No oil | 10.85 |
|  | With oil | 24.60 | With oil | 5.80 |
|  | Change | −53.33% | Change | −87.07% |
| **Average value** | No oil | 2.88 | No oil | 0.45 |
|  | With oil | 1.85 | With oil | 0.32 |
|  | Change | −55.78% | Change | −39.05% |
| **Kurtosisvalue** | No oil | 7.46 | No oil | 25.12 |
|  | With oil | 7.63 | With oil | 16.03 |
|  | Change | 2.18% | Change | −56.73% |
| **Skewness Value** | No oil | 2.54 | No oil | 3.97 |
|  | With oil | 2.56 | With oil | 3.14 |
|  | Change | 0.98% | Change | −26.39% |

Figure 17 and Table 4 present the results for the displacement and acceleration of the reaction when the Taft seismic waves were used. As was the case of excitation using EI Centro NS seismic waves, the response displacement and acceleration are smaller for hydraulic oil; the results of the spectral analysis show that the vibration response is smaller over a wide range of frequencies.

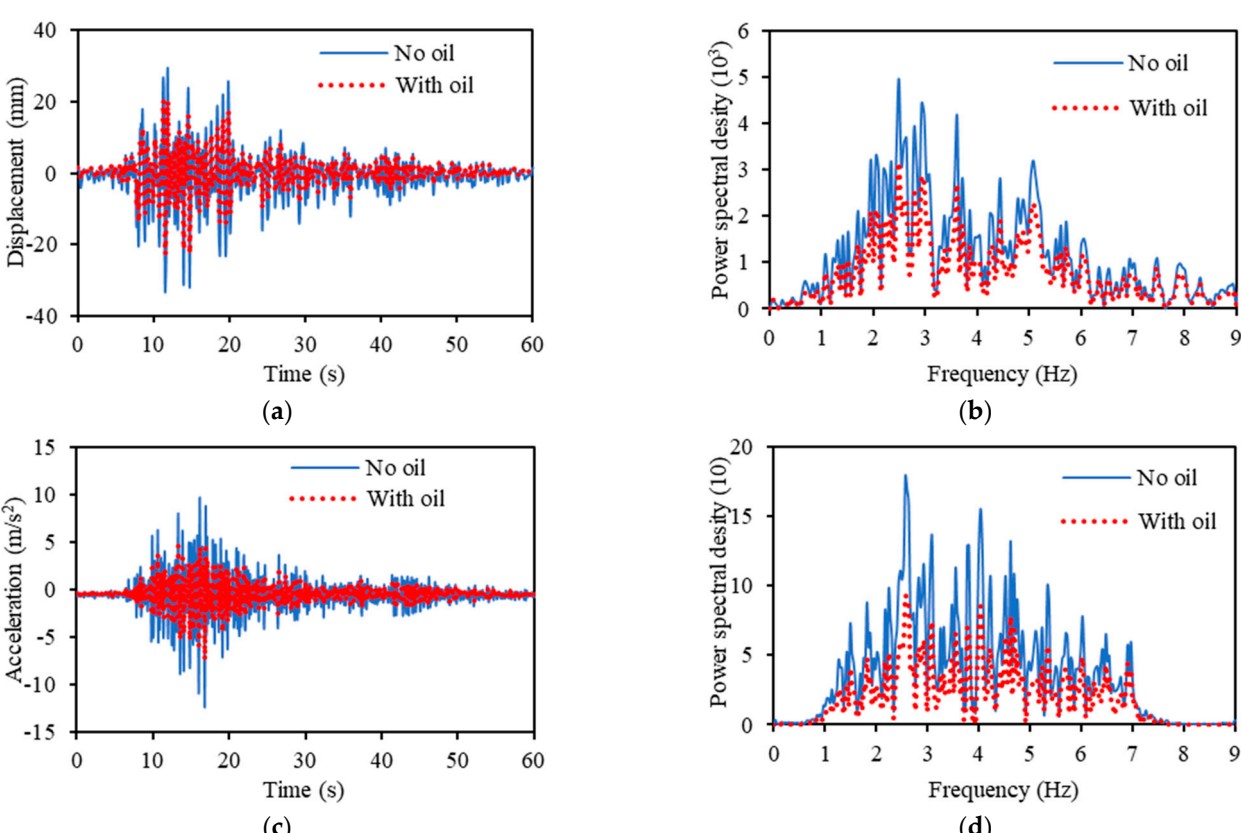

**Figure 17.** Measurement results when excited by Taft seismic wave. (**a**) Response displacement. (**b**) Spectral distribution of displacement. (**c**) Response acceleration. (**d**) Spectral distribution of acceleration.

**Table 4.** Comparison of measurement results when excited by Taft seismic wave.

| | | Displacement | | Acceleration |
|---|---|---|---|---|
| | | | | |
| **Maximum value** | No oil | 33.41 | No oil | 12.04 |
| | With oil | 22.73 | With oil | 7.33 |
| | Change | −46.99% | Change | −64.26% |
| **Average value** | No oil | 3.99 | No oil | 0.61 |
| | With oil | 2.64 | With oil | 0.38 |
| | Change | −50.88% | Change | −59.02% |
| **Kurtosisvalue** | No oil | 1.62 | No oil | 18.52 |
| | With oil | 1.84 | With oil | 17.33 |
| | Change | −11.80% | Change | −6.91% |
| **Skewness Value** | No oil | 1.42 | No oil | 3.61 |
| | **With oil** | 1.45 | **With oil** | 3.38 |
| | **Change** | −2.13% | **Change** | −6.79% |

Table 4 shows that the maximum value of the response displacement was reduced by 46.99%, and the average was reduced by 50.88%. The maximum value of the response acceleration was reduced by 64.26%, and the average was reduced by 59.02%. The kurtosis value of the response displacement was reduced by 11.80%, and the skewness value was reduced by 2.13%. The kurtosis value of the response acceleration was reduced by 6.91%, and the skewness value was reduced by 6.79%.

To assess the complex vibration damping effect quantitatively, the standard deviation *S* indicated in Equation (32) was used as the evaluation reference value.

$$S = \sqrt{\frac{1}{N}\sum_{i=1}^{N}(x_i - \overline{x})^2} \, , \tag{32}$$

where $x_i$ is the measured displacement or acceleration of the vibration response, $\overline{x}$ represents the average value of the measured $x_i$, and $N$ is the number of sample points in the measurement experiment.

The results derived by substituting the measurement results in Figures 16 and 17 into Equation (32) are shown in Figure 18. Figure 18a illustrates the results of the measurement with the EI Centro NS seismic wave, and Figure 18b shows the results of the measurement with the Taft seismic wave.

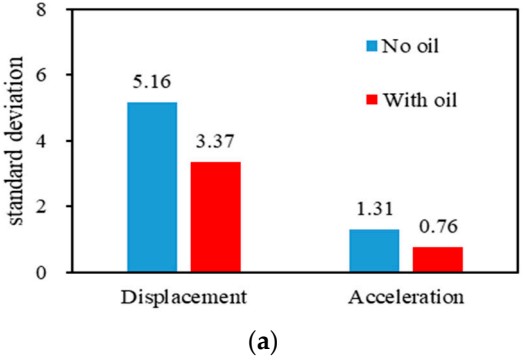

(**a**)

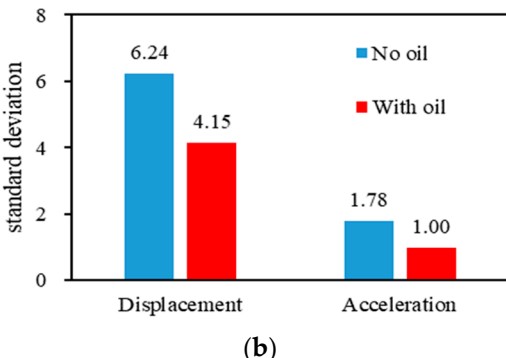

(**b**)

**Figure 18.** Comparing standard deviations of displacement and acceleration. (**a**) EI Centro NS seismic waves. (**b**) Taft seismic waves.

Figure 18a shows that when shaken by the EI Centro NS seismic wave, the standard deviation of the response displacement decreased by 34.83% from 5.16 to 3.37; the standard deviation of the response acceleration decreased by 42.35% from 1.31 to 0.76. Figure 18b indicates that when shaken by the Taft seismic wave, the standard deviation of the response displacement decreased by 33.58% from 6.24 to 4.31 and the standard deviation of the response acceleration decreased by 43.82% from 1.78 to 1.00. The measurement results indicate that the origami hydraulic dampers provide effective damping.

### 3.2. Effects of Orifice Hole

The size of the damping hole through which the hydraulic fluid flows significantly affects the damping effect. The response displacement and response acceleration for different orifice hole diameters $d_1$ were tested in vibration experiments, as shown in Figure 8.

Figures 19 and 20 show the results of the vibration experiments conducted with the orifice hole diameter divided into two scenarios when adding hydraulic oil with a normal diameter of 12 mm and a small diameter of 8 mm.

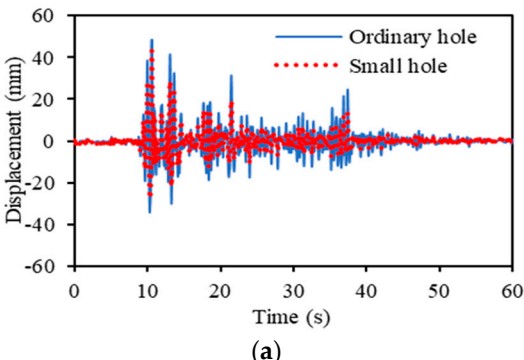
(**a**)

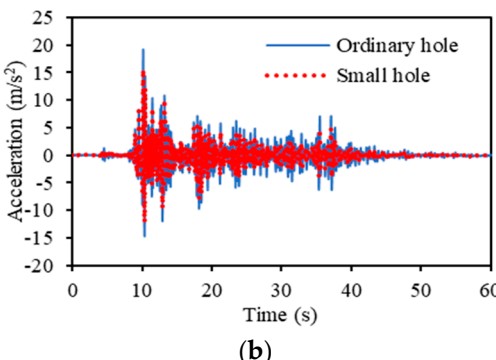
(**b**)

**Figure 19.** Effect of attenuation hole diameter when excited by EI Centro NS seismic wave. (**a**) Response displacement. (**b**) Response acceleration.

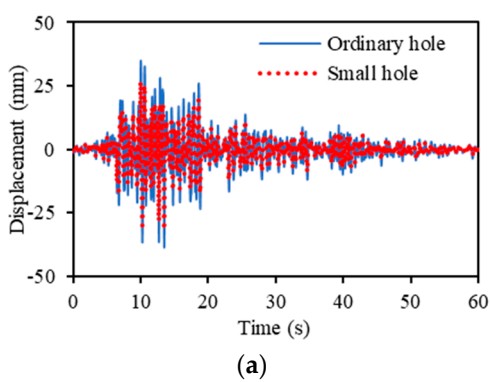
(**a**)

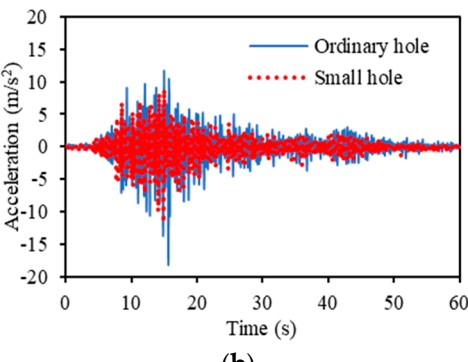
(**b**)

**Figure 20.** Effect of attenuation hole diameter when excited by Taft seismic wave. (**a**) Response displacement. (**b**) Response acceleration.

Figure 19 shows the results of the response displacement and response acceleration when shaken by the EI Centro NS seismic waves. It can be observed that the response displacement and response acceleration decreased as the size of the orifice hole decreased. The standard deviation of the response displacement decreased by 45.42% from 6.80 to 4.67, and the standard deviation of the response acceleration decreased by 33.37% from 2.07 to 1.55. Similarly, Figure 20 shows the measured results of the response displacement and response acceleration when subjected to Taft seismic wave excitation. By decreasing the size of the orifice hole, the standard deviation of the response displacement decreased by

31.28% from 6.70 to 5.11, and the standard deviation of the response acceleration decreased by 36.60% from 2.19 to 1.60.

Therefore, it was established that the damping effect of the origami hydraulic damper increases as the orifice hole diameter decreases.

### 3.3. Effect of Fluid Type

The characteristics of the fluid used in an origami hydraulic damper may affect the damping effect. An experiment was set up to investigate this issue by placing a hydraulic fluid and water in the origami damper of the setup shown in Figure 12. The response displacement and response acceleration in the experiment were examined.

Figures 21 and 22 show the results acquired from the shaking tests conducted using hydraulic oil and water, respectively.

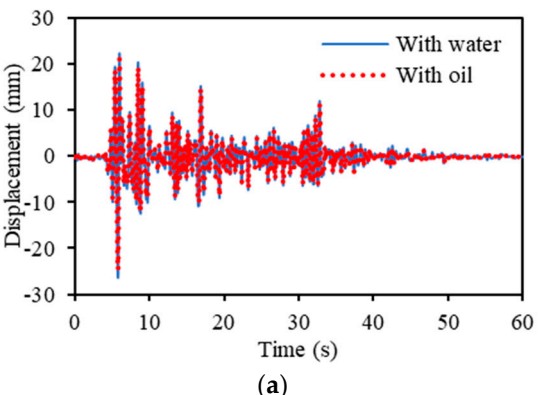
(a)

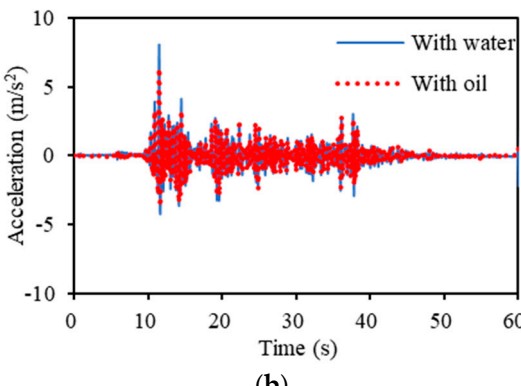
(b)

**Figure 21.** Effect of type of liquid when excited by EI Centro NS seismic wave. (**a**) Response displacement. (**b**) Response acceleration.

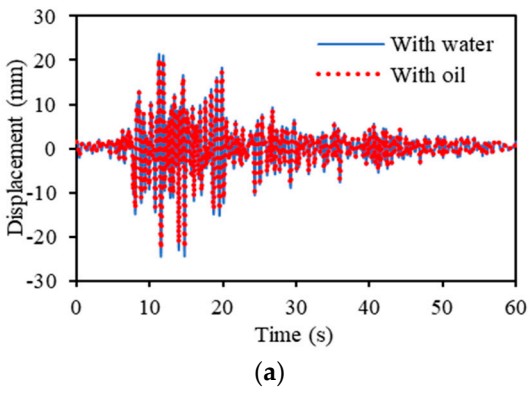
(a)

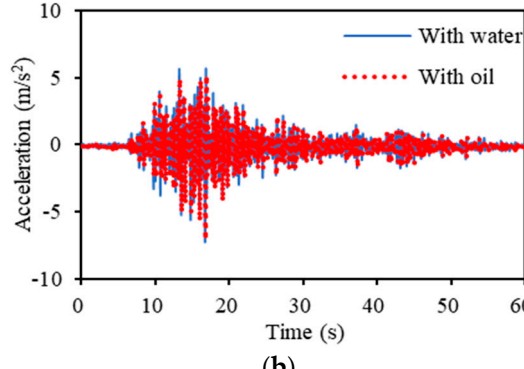
(b)

**Figure 22.** Effect of type of liquid when excited by Taft seismic wave. (**a**) Response displacement. (**b**) Response acceleration.

Figure 21 illustrates the measured results of the response displacement and response acceleration when shaken by the EI Centro NS seismic wave. A trend of smaller response displacement and response acceleration can be observed by adding hydraulic oil rather than water. The standard deviation of the response displacement decreased by 6.82%, falling from 3.59 to 3.37, and the standard deviation of the response acceleration decreased by 24.48%, falling from 0.71 to 0.57. Similarly, Figure 22 shows the measured response displacement and response acceleration when shaken by Taft seismic waves. The standard deviation of the response displacement decreased by 6.96%, from 4.44 to 4.15; the standard deviation of the response acceleration decreased by 23.52%, from 0.91 to 0.74, when hydraulic oil was used compared to water.

As the viscosity of hydraulic oil exceeds that of water, it is probable that the damping effect of the origami damper is higher with the addition of hydraulic oil than with water. However, in comparison with the results of the previous section, the effect of adjusting the fluid type tends to be smaller than the effect of adjusting the orifice hole size in terms of the effect of the origami damper on the damping effect.

## 4. Conclusions

This study proposed an inverted spiral-type origami hydraulic damper to replace the conventional cylindrical hydraulic damper. The geometric conditions and mechanical properties of the origami hydraulic damper were examined. A validation test setup using the origami hydraulic damper was fabricated to perform practical vibration tests. The following conclusions were drawn from the results:

(1) The free-folding conditions of the inverted spiral origami structure for hydraulic dampers revealed that the necessary conditions are that the inclination angles of the folding lines on the sidewalls should be $\alpha = \pi/n$ and $\pi/2n \leq \beta \leq \pi/n$.

(2) Theoretical study on the damping force of the origami hydraulic damper showed that the damping force of the origami hydraulic damper is proportional to the square of the velocity of motion. An equation for the damping coefficient of the origami hydraulic damper was derived. In addition, the nonlinear equations of motion, including mass block, elastic spring, and a damping element, were formulated as vibration-damping objects. A vibration analysis method employing the Range–Kutta method was devised.

(3) Detailed studies on the damping performance of the origami hydraulic damper were conducted using experimental equipment. Using seismic waves as a test input, it was evident that the proposed origami hydraulic damper provided a dependable damping effect under complex excitation conditions. It could be used as a vibration-damping device.

(4) This experimental study on the effect of the major configuration parameters of the origami hydraulic damper on the damping performance indicates that the size of the hole at the end of the origami hydraulic damper is rather significant, depending on the fluid characteristics used in the damper.

However, compared to the conventional cylinder-type hydraulic damper, the origami hydraulic damper was constructed with a thin origami structure, so it was difficult to apply to the problem of large vibration loads such as heavy trucks.

In future work, we will apply the origami hydraulic damper proposed in this research to problems such as the problem of narrow installation space, the problem of weight reduction, and the problem of unstable vibration direction.

**Author Contributions:** Writing—original draft preparation, J.G.; writing—review and editing, J.G. and X.Z.; data curation, J.Z.; investigation, N.G.; software, W.Z.; conceptualization, W.Z. and X.Z.; methodology, J.G. and J.Z.; validation, N.G. All authors have read and agreed to the published version of the manuscript.

**Funding:** This research received no external funding.

**Data Availability Statement:** Data available in a publicly accessible repository.

**Conflicts of Interest:** The authors declare no conflict of interest.

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
