# Peer review of "Study on Hydraulic Dampers Using a Foldable Inverted Spiral Origami Structure"

_vibration, doi:10.3390/vibration5040042_

Round 1

Reviewer 1 Report

The work is very good, I suggest in the conclusions a better and extensive highlighting of the practical applications.

Reviewer 2 Report

This work presents a novel axially free-folding damper design based on the inverted spiral origami structure. The key advantage of this design is its essentially unlimited compression ratio, which contrasts the classic tubular damper (which has a 50% compression ratio to the maximum).  It is interesting to see the popular and widely applied concept of origami applied to a traditional fluid mechanics application.

Major comments

What exactly was validated by the agreement in Fig 13, that the damping effect of the origami damper is nonlinear? If this is the point that the authors were making, then I would question whether the experimental design can serve this purpose. Because the system is being driven by an external force that appears rather strong, any damping effect could be overshadowed by the force: in fact not much damping effect is visible in Fig. 13. In addition, both linear and nonlinear dampers appear in the setup, making it hard to distinguish the nonlinear damping effect. 

In the reviewer’s opinion, a much better way of validating the quadratic damping effect is to have a free oscillation system using just the origami damper, a spring, and an initial displacement. The displacement curve then should resemble something like exp(-a t) cos(b t) from undergraduate physics text. This result not only provides strong evidence of the damping effect but can also quantify the damping coefficient, which can be compared with the phenomenological model presented in eqn. 15.

Minor comments

  • fig. 3 left does not have the label "a"

  • tab. 1, the left column should also list the symbols used in the equations.

  • line 238: with "(velocity -> zero velocity)" as the center point?

  • line 240: why it can be assumed linear when it is nonlinear?

  • eqn. 17, what are the values of m, c, K?

  • for people without a background in origami, it is unclear why eqn. 1 is mandatory for a flat fold. One or two sentences' elaboration could be helpful. A quick search indicates this is Kawasaki's theorem. If it is, please provide references for this work.

Reviewer 3 Report

I liked this article, and I would say this is one of the best studies I’ve encountered. First of all, congratulations for putting up such a fantastic idea and investigation. Following are my comments.

1.      Can you add code for the Runge-Kutta method in the annexure? Also, a flowchart of the Runge-Kutta method can be added to the paper so that readers would understand the methodology easily.

2.      Comment on the sustainability of spiral origami structure. Justify material selection.

3.      It is good to add structural or modal analysis of the structure to justify the design of the structure.

4.      In table 4, why were only maximum, average value, and standard deviation compared? Try calculating other features such as kurtosis, skewness, etc.

5.      What’s the scope of a MEMS-based accelerometer integrated with an open source controller such as Arduino etc. for measurement of vibrations? Refer to the articles, ‘Novel Machine Health Monitoring System,’ ‘Application of Machine Learning for Tool Condition Monitoring in Turning,’ and ‘Tyre Pressure Supervision of Two Wheeler Using Machine Learning' for an idea of developing in-house low-cost DAQ. You may suggest this in future scope.

6.      In continuation, try connecting your work to the open-design movement reviewed in the paper ‘Overview of contemporary systems driven by open-design movement’. This would open new opportunities for authors and readers as well.

7.      Try eliminating lines such as ‘The solid blue line represents the measurement results with water, and the dotted red line indicates the measurement results with hydraulic oil’ as it is redundant.

8.      Can you establish the following statement mathematically so that designers can use it for developing scaled-up or down models in the near future? The statement is “However, in comparison with the results of the previous section, the effect of adjusting the fluid type tends to be smaller than the effect of adjusting the orifice hole size in terms of the effect of the origami damper on the damping effect.” OR “Therefore, it has been established that the damping effect of the origami hydraulic damper increases as the orifice hole diameter decreases.”

9.      What’s the limitation of this work? Please add it.

The article is lovely but needs some revision as suggested herein for final recommendation. All the best!

Round 2

Reviewer 2 Report

The authors have addressed all my comments and I find the presentation now clearer. I would recommend the publication of the article in its present form.